# Psychosocial and Physical Predictors of Stress in University Students during the COVID-19 Pandemic: An Observational Study

**DOI:** 10.3390/healthcare10050786

**Published:** 2022-04-23

**Authors:** Nutsupa Ubolnuar, Nongnuch Luangpon, Krittipat Pitchayadejanant, Sirirat Kiatkulanusorn

**Affiliations:** 1Department of Physical Therapy, Faculty of Allied Health Sciences, Burapha University, Chonburi 20131, Thailand; nutsupa.ub@go.buu.ac.th (N.U.); nongnuchl@go.buu.ac.th (N.L.); 2Burapha University International College, Burapha University, Chonburi 20131, Thailand; krittipat@go.buu.ac.th

**Keywords:** COVID-19, physical factors, psychosocial factors, stress, university students

## Abstract

Currently, university students are at a high risk of stress due to university adjustment, educational interruption, and alterations in daily life because of the COVID-19 pandemic. This study examined the relationship of psychosocial and physical factors with stress in university students during the pandemic. Demographic, psychosocial, physical, and self-perceived stress level information were obtained from 409 Thailand university students. A multiple regression analysis was performed, with stress level as the dependent variable and gender, age, study period, study program, social support, self-esteem, health literacy, health behavior score, sedentary behavior, and physical activity (PA) as independent variables. Most participants had moderate stress levels (68.9%), high self-esteem (83.9%) and social support (66.5%), fair health literacy (41.1%) and health behavior (32%), sedentary lifestyle (85.3%), and PA-levels lower than 600 min per week (57.46%). The regression analysis showed that 45.7% of the variability in stress level was predicted by self-esteem, study period, social support, travel domain of PA, and health behavior. COVID-19 and the attendant restrictions resulted in moderate levels of stress in Thailand university students. High self-esteem, long duration of study, great social support, and having healthy behavior may contribute to the prevention of stress in this population.

## 1. Introduction

COVID-19 has been a global pandemic since December 2019 [1]. It has affected the economic and political climate and the public health of the world population [2,3]. The World Health Organization (WHO) has advised maintaining a distance of at least 1 m from others, avoiding public areas or crowded places, and staying at home as much as possible. Lockdown measures have been implemented in organizations, universities, schools, and public areas, to contain and limit the spread of COVID-19 [3].

In Thailand, the first case of COVID-19 was reported in January 2020. The number of cases increased drastically in March 2020, partially due to the emergence of nightclub and boxing stadium clusters. Following the WHO guidelines, the Thai government implemented lockdown measures that included a curfew between 10 p.m. and 4 a.m., shutting down nonessential businesses and organizations to keep workers at home, and closing public areas and crowded places [4]. Both international and domestic travel were restricted to limit movement and social interaction [3].

The spread of COVID-19 has directly affected people’s lifestyles. Social and physical distancing measures, along with the requirement to wear a mask in public areas were implemented to reduce infection [5]. However, daily physical activities (PAs), such as working, exercising, and outdoor pursuits, have all been limited by the restrictions [1]. The changes to daily life have affected the physical and psychological health of the world population. Moreover, the ongoing exposure to information about COVID-19, including uncertainties about health and the future; irritability due to work, study, or project disruption; reduced incomes; and political and economic issues, could exacerbate stress, depression, and other psychological problems [5]. Optimal health literacy and behavior are required to mitigate the detrimental effects of a pandemic [6]. Despite its importance to health and well-being, health literacy remains an area of inquiry, which is often neglected by research.

Due to the lockdown measure, universities have postponed classes or changed from onsite to online education [3,7]. These changes result in increased stress levels among university students [8]. This may cause some students to fail in their learning program, with negative effects emotionally and academically on their future careers [9]. The lockdown measures may also affect family incomes [10] and the social life of students, such as a reduction or the absence of recreational activities, sports, or PA [8,9]. Low PA has been reported among university students in different countries [11]. There has also been an increase in students’ social media use during COVID-19, and this has had negative effects on mental health [10]. Moreover, most university students are young adults, some of whom may lack the adequate skills to adjust during the COVID-19 pandemic [8]. Thus, there have been negative effects on both psychological and physical health in university students during the COVID-19 pandemic [9].

Apart from PA, psychosocial factors, including self-esteem and social support, have been affected by COVID-19 [12,13]. Self-esteem plays an important role in the mediation of stress-related biological processes and is associated with subjective well-being, effective biological regulation, and physical health [14]. Arsandaux et al. [12] found that university students were at higher risk of mental health disturbances, including reductions in self-esteem during lockdown than nonstudents [12]. Social support is known to increase resilience to stress, thereby improving mental and physical health [15]. Adequate coping strategies and social support are significantly correlated with lower psychological distress in university students during COVID-19 [13]. Friedlander et al. studied the predictors of adjustment to university in Canadian students and demonstrated that social support, self-esteem, and stress level are important predictors of adjustment to university among first-year undergraduate students [16].

COVID-19 increases stress in students, activating physiological, emotional, and behavioral responses [14]. Moreover, different situations, areas and occupations could influence the impact of the pandemic on individuals and population categories. There is no study on the impact of the pandemic, especially the first wave, on Thailand university students. Due to the novelty of the pandemic and accompanying restrictions, Thai university students are at high risk of stress because of learning adjustment, educational interruption, financial concerns, family problems, alterations in daily life, social isolation, etc This study investigated the psychosocial and physical factors influencing the stress level in Thailand university students during the COVID-19 pandemic. Understanding these factors can be beneficial in the development of strategies aimed at stress reduction and prevention of psychological disorders.

## 2. Materials and Methods

### 2.1. Study Design

This was a cross-sectional, observational, analytical, and descriptive study. Participants were recruited using a stratified random sampling method, and they submitted online consent forms before participation. Neither patients nor the public were involved in the design, conduct, report, or dissemination of this research plans.

### 2.2. Participants

The sample size was calculated using the formula by Taro Yamane [17]. The population of students was 25,743. With a significance level of 0.05, the minimum valid sample size was calculated to be 395 students. Undertaking a proportional stratified sampling of students from each faculty, 409 students at Burapha University were recruited for this study.

For confounders’ control, inclusion criteria were determined that the student was more than 18 years old, studying at Burapha University’s Bangsaen campus, and accessible on electronic devices to complete the online questionnaires. Participants were excluded if they were unable to read or understand the Thai language because the questionnaire was written in Thai. The confounders in the study were age, specific location, and nationality.

Participants were requested to participate in the study by responding to an online invitation posted on the official social media handles of each faculty’s student affairs (Facebook and LINE Application). The participants completed an anonymized online questionnaire that was created using Google Forms. Completed online questionnaires were collected and analyzed as participants of the study.

### 2.3. Data Collection and Instrument

This comprised six sections: demographic data, PA, social support, self-esteem, health literacy and health behavior, and perceived stress. Data were collected in October 2020 using a self-administered questionnaire. The validity and reliability of the instrument was assured through wide consultations with relevant academicians, a review of the literature, and adaptation of validated instruments. The questionnaire comprised of six sections: (1) sociodemographic characteristics, (2) PA, (3) social support, (4) self-esteem, (5) health literacy and health behavior, and (6) perceived stress. The participants were informed to rest for 5 min after completing each section of the questionnaire. 

PA was assessed using the Global Physical Activity Questionnaire (GPAQ) v.2 [18]. The Thai version of GPAQ was published by Thailand’s Ministry of Public Health. A previous study showed a reliability of 0.67- 0.73 and a moderate (0.45) to strong (0.65) validity [19]. Information was collected on three domains of PA: activity at work, travel, and recreation. The energy used in PA from each domain is calculated as metabolic equivalent minutes per week. Total PA is calculated on the basis of the overall score from all domains. The level of PA was considered in relation to that recommended by WHO (≥600 min/week). An additional section of the questionnaire that we classed as the sedentary behavior domain was also collected. The level of sedentary behavior was categorized into nonsedentary, sometimes sedentary, and always sedentary [20].

Levels of social support were measured using the Thai version of the revised Multidimensional Scale of Perceived Social Support. The questionnaire consists of 12 items that ask about the support from one’s family, friends, and significant others. Answers were presented on a 7-point Likert scale. The total score was the sum of the ratings for the 12 items. We classified these as high (scores of 61–84), moderate (scores of 37–60), and low support (scores of 12–36). The reliability of this questionnaire was between 0.86 and 0.92 [21].

Self-esteem was measured using the Thai version of the Rosenberg Self-Esteem Scale. The questionnaire consists of 10 items measured on a Likert scale of 0–4 points. Total possible scores ranged from 10 to 40. The levels of self-esteem were classified as high (scores > 32), moderate (scores of 26–32), and low self-esteem (scores < 26). This questionnaire had a strong reliability of 0.86 [22].

Health literacy was assessed using the health literacy and health behaviors 3E 2S evaluation of the Health Education Division of Thailand’s Ministry of Public Health. This questionnaire consists of six sections, including (i) demographic data (12 items), (ii) knowledge and understanding of health (6 items), (iii) accessibility of health information (10 items), (iv) decision making based on perceived health information (3 items), (v) health behavior (6 items), and (vi) health information (8 items). The Kuder–Richardson 20 score for this evaluation was 0.516 and it had a Cronbach’s alpha coefficient of 0.602–0.788.

Stress levels were measured using the Thai version of the 10-item Perceived Stress Scale. The questionnaire consists of 10 items measured on a Likert scale of 0–4 points. Stress was classified into three levels: high stress (scores 0–13), moderate stress (scores 14–26), and low stress (scores 27–40). The reliability of this questionnaire was 0.85 [23].

### 2.4. Statistical Analysis

Data were analyzed using SPSS software. Stress level (ratio level) was our dependent variable (DV), whereas our independent variables (IVs) were gender, age, study period, study program, self-esteem, social support, health literacy, health behavior, sedentary behavior, and PA (including the work, travel, and recreational domains). We performed ordinary multiple regression analysis because of its suitability to our data set. Our study’s nominal data (gender, study program (health sciences vs. nonhealth sciences), and sedentary behavior (nonsedentary vs. sedentary behavior)) were recoded and transformed into dummy variables. To maintain the reliability of the instrument and reduce complexity in applying the model, the scale was not adjusted in this study. Due to concerns about the variability of different weightage in the Likert scale, the unstandardized coefficient was reported to predict the stress level. The model can be implemented into unobserved observation without transforming the data. Additionally, the standardized coefficient was reported to account for the comparison of the strength of factors affecting stress levels. Furthermore, assumptions of ordinary regression analysis were evaluated. The Kolmogorov–Smirnov test and scatter plots were used to test the normality and homoscedasticity of residuals. Autocorrelation and the multicollinearity of the IVs were tested using Durbin–Watson statistic, tolerance test, and variance inflation factor. Pearson’s product–moment correlation coefficient was used to examine the relationship between the stress level and demographic variables, as well as between the stress level and DVs. The significance level in the study was 0.05.

## 3. Results

A total of 523 questionnaires were retrieved from the students; however, 409 completed questionnaires were analyzed and included in this result. Participants’ characteristics are presented in Table 1. The majority of the participants were females (n=297,  72.62%) and in nonhealth sciences programs (n=334,  81.66%), whereas approximately 40% (n=161,  39.36%) were in the fourth academic year.

The correlations between the participants’ characteristics and stress levels are presented in Table 1. Age (r=−0.221,  p<0.001) and study period (r=−0.252,  p<0.001) had a weak, significant negative correlation with stress levels.

The correlations between independent variables and stress levels are presented in Table 2. Self-esteem (r=−0.625,  p<0.001) and social support (r=−0.425,  p<0.001) had a moderate, significant negative correlation with stress levels. However, health literacy (r=−0.140,  p=0.004) and health behavior (r=−0.241,  p<0.001) had a weak, significant negative correlation with stress level. Activity domain (r=0.120,  p=0.015) and travel domain (r=0.137,  p<0.005) of PA had a weak, significant positive correlation with stress level. [24]. The remaining variables were not significant.

The results of ordinary multiple regression analysis are presented in Table 3. The variables that were not significant in correlation were excluded from this analysis. The stepwise method iteratively examines each significant independent variable in the model. The regression model comprised five significant independent variables (F = 67.90, *p* < 0.001): self-esteem, study duration, social support, the travel domain of PA, and health behavior. Overall, 45.7% of the variability in stress level was predicted by these five independent variables. In addition, the Durbin-Watson test of autocorrelation (n=409,  k=5,  α=0.05,  du=1.794) result was 1.851. Based on the acceptable range of no evidence of autocorrelation (1.794 and 2.206), our result showed that there is no evidence of autocorrelation. The residuals presented normality and homoscedasticity.

In this study, multicollinearity was also investigated by variable inflation factors (VIF), which should be close to 1 and lower than 5. As shown in Table 4, the VIF value for each variable is between 1.024 and 1.409. This finding shows that our study had no multicollinearity issue.

Table 4 shows the unstandardized regression coefficients (B), the standardized regression coefficient (β), and the semi-partial correlations (Sr^2^). For the final model, five IVs contributed significantly to stress level prediction: self-esteem (B=−0.680,β=−0.497,p<0.001 ***), study period (B=−1.064,β=−0.168,p<0.001 ***), social support (B=−0.084, β=−0.158,p<0.001 ***), the travel domain of PA (B=0.002,β=0.108,p= 0.004**), and health behavior (B=−0.199,β=−0.093,p<0.017 *). Hence, the equation model for predicting stress levels was derived for these variables: 

Predicted stress level = 50.21 − 0.68 (self-esteem) − 1.064 (study period) − 0.084 (social support) + 0.002 (travel domain of PA) − 0.199 (health behavior)

## 4. Discussion

This study assessed the relationship of self-esteem, social support, health literacy and health behavior, and PA with stress in Thailand university students during the first wave of the COVID-19 pandemic. Our findings indicated that the combined influence of psychosocial and physical factors predicted 45.7% of the stress levels of Thailand students during the COVID-19 lockdown. The regression equation produced a moderate fit with the data (R^2^ = 0.457), with all predictors, including self-esteem, study duration, social support, health behavior, and the travel domain of PA.

To our knowledge, this study is the first to examine the combination of psychosocial and physical factors that predict stress levels in university students during COVID-19. Previous studies investigated the effects of COVID-19 on physical [1,11] or psychosocial [7,25] or mental health factors among university students [8,9]. Saleh et al. reported that a combination of psychological variables, including life satisfaction, self-esteem, optimism, self-efficacy, and psychological distress, could account for 57.89% of stress levels in French college students [26]. However, their research data were collected in 2017; therefore, their data cannot be compared with the data collected in the COVID-19 pandemic [26]. In 2020, Flesia et al. investigated stress among the Italian normative population during COVID-19 [27]. Sociodemographic variables could predict only 10.1% of stress, whereas the combination of sociodemographic and stable psychological traits could predict 35.6% [27]. However, both are low data fit models [24].

We found that the main predictors of university student stress during COVID-19 are self-esteem and social support. Consistent with previous studies, self-esteem and social support are important mediators of stress and its pathological effects [28,29]. It is possible that low self-esteem and low social support also increase the stress-induced cortisol release by modulating the regulation of the hypothalamic–pituitary–adrenal (HPA) axis and the noradrenergic system. Low self-esteem increases cortisol levels by stimulating diurnal cortisol secretion, and also increases disturbances of the HPA axis, which are associated with stress. Self-esteem affects people’s reactions and coping mechanisms to stressful events. Simultaneously, stressful events negatively affect self-esteem [14]. Increases in cortisol levels is usually associated with stress. Cortisol levels stimulate fat and carbohydrate metabolism but increases appetite. Additionally, increased cortisol levels can cause cravings for sweet, fatty, and salty foods, which can lead to weight gain if uncontrolled [30]. Herman and coworkers [31] found multiple and overlapping mechanisms to deal with both acute and chronic stress and named the reaction of stressors to homeostasis as “stress response,” wherein the HPA axis secretes the first hormonal response to homeostatic challenge when stimulated. Predominantly, few HPA axis changes are engendered by all varieties of stressor and are a hallmark of the physiological reaction to stress. Proper control of the stress response is of critical importance. Because of the long-term approaches in stress situations due to the COVID-19 pandemic, it can be implied that the pandemic caused prolonged HPA axis activation, which was linked with numerous physiological and psychological disease states. Social support was another predictor of stress level during the COVID-19 pandemic. A high level of social support can enhance resilience to stress, help protect against developing trauma-related psychopathology, and decrease the functional consequences of trauma-induced disorders [15]. The results of social support in human studies showed that low social support is related to physiological and neuroendocrine markers of intense stress response in laboratory stressors. A neuroendocrine that has been detected in humans during the stress response is oxytocin. The study design’s laboratory stressor test, by simulation of acting in a public situation with negative feedback led to anxiety and salivary cortisol release. This result showed that both oxytocin and social support reduced anxiety in healthy men. Therefore, they propose that oxytocin promotes social behavior and reduces HPA axis reactivity to stress [32]. These findings are consistent with the results of a study conducted by Steptoe et al. [33], who reported an overall increased noradrenergic and HPA reactivity in lonely individuals.

An interesting finding of this study was that, during the COVID-19 pandemic, there was an increase in travel activity among students, and this was positively associated with high levels of stress. During the first wave of the COVID-19 pandemic in Thailand, most universities adopted online learning from home for their students. They have the least medical risks, allowing them to go outside to obtain necessities for a living in order to support their family. In normal situations, individuals who are regularly physically active have lower levels of stress than those who are less active [1]. The higher stress level during travel is likely to be due to feared exposure to COVID-19. Although the students will have been wearing masks and social distancing, there is still a risk of COVID-19 infection during travel to public areas. Moreover, the low confidence in pandemic control and the lack of resources for fighting COVID-19 might have further affected stress levels in the circumstances, especially the first wave of the pandemic [25,34].

High levels of stress were significantly correlated with young ages, short study duration, poor health literacy, and poor health behavior. Students with better health literacy can more actively adapt their health behavior, especially during COVID-19 [35]. Rababah et al. demonstrated that improving students’ health literacy and health behavior directly affects psychological disturbance, reducing levels of stress and improving quality of life [36]. In our study, most of the students had high levels of social support and self-esteem and fair health literacy and behavior. Although younger students who had been at the university for a shorter time were vulnerable to higher levels of stress, most of them may be able to handle the situation due to their high levels of social support and self-esteem [29].

The majority of the students in our study reported moderate levels of stress, an increase in sedentary lifestyle, and lower levels of PA as a result of COVID-19. The data collection period was between the first and second outbreaks of COVID-19 in Thailand. At that time, COVID-19 was new, and due to lockdown measures, it directly affected students’ lifestyles [37]. Students had to avoid public areas, including their classes and activities [3,7]. Most showed fair health literacy and behavior, possibly due to the lack of information about COVID-19 and pandemic control in Thailand. The high levels of self-esteem and social support might be explained by the family structure in Thailand, where teenagers and young adults usually live with their parents until they graduate or marry. Hertog and Kan [38] found that people of working age also sometimes choose to live with their parents for parental support or childcare. This family support might contribute to students’ levels of self-esteem and social support during COVID-19.

This study provided knowledge on stress levels related to psychosocial and physical factors during the first wave of the COVID-19 pandemic. Our result can be applied to the proper management of new and life-threatening pandemics, which impact human well-being.

Our study has some limitations. First, it was a cross-sectional online survey that used a questionnaire to elicit information from participants. Therefore, there was no control group to determine a cause-and-effect relationship. In addition, the responses were mainly subjective, the questionnaire was subject to varied interpretation and understanding, errors in responses, and mental fatigue by participants. Second, depression, anxiety, and posttraumatic stress disorders, which have been linked to the COVID-19 pandemic, were not measured [37]. Future studies on these psychological disorders and other complications associated with the COVID-19 pandemic are needed.

## 5. Conclusions

The first wave of the COVID-19 pandemic negatively impacted the stress levels of university students in Thailand. The majority of them presented with moderate stress levels. The combination of psychosocial and physical factors, including self-esteem, study duration, social support, health behavior, and the travel domain of PA predicted their level of stress during the pandemic. Higher self-esteem and social support were the factors that significantly helped students cope with stress. Moreover, mindfulness meditation can help people relax and regulate their emotions via upregulation and calming. Yoga, which includes cleansing techniques, physical postures, breathing exercises, and relaxation techniques, should be recommended to those who want to reduce stress and improve their self-esteem. Additionally, online platforms can be used to improve social support during pandemics by enhancing students’ attraction and their involvement in studies and other activities. Universities may play an essential role in providing online counseling and workshops to students. However, students experienced higher stress with traveling within public spaces. Further studies should investigate the long-term effect of COVID-19 pandemic on stress levels and the stress management strategies among specific categories of university students. Furthermore, experimental studies are needed to confirm the positive effect of self-esteem and social support on stress levels during pandemics or life-threatening situations. 

## Figures and Tables

**Table 1 healthcare-10-00786-t001:** Demographic data of participants and correlations between participant characteristics and stress level.

Participants Characteristics	Frequency	Percentage	Total		Correlation with the Level of Stress	
			Mean	SD	r	*p*-Value
**Age (year)**			20.95	1.24	−0.221	<0.001 ***
**Gender**						
- Male	112	27.38			−0.078	0.113
- Female	297	72.62				
**Study period (year)**			2.126	1.04	−0.252	<0.001 ***
**Year of study**						
- First academic year	61	14.91				
- Second academic year	56	13.69				
- Third academic year	131	32.03				
- Fourth academic year	161	39.36				
**Study program** **(health: nonhealth)**					0.046	0.356
- Health sciences	75	18.34				
- Other	334	81.66				
**Relationship status**					−0.019	0.708
- Single	405	99.02				
- In a relationship	4	0.98				

*** *p* < 0.001.

**Table 2 healthcare-10-00786-t002:** Correlations between stress level and self-esteem, social support, health literacy, health behavior, physical activity, and sedentary behavior.

Variables	Frequency	Percentage	Total		Correlation with the Level of Stress	
			Mean	SD	r	*p*-Value
**Stress score**			17.82	6.61		
- Low perceived stress	96	23.5				
- Moderate perceived stress	282	68.9				
- High perceived stress	31	7.6				
**Self-esteem**			30.68	4.82	−0.625	<0.001 ***
- Low self-esteem	2	0.5				
- Normal self-esteem	64	15.6				
- High self-esteem	343	83.9				
**Social support**			63.50	12.36	−0.425	<0.001 ***
- Low perceived social support	9	2.2				
- Moderate perceived social support	128	31.3				
- High perceived social support	272	66.5				
**Health literacy**			44.72	6.22	−0.14	0.004 **
- Poor health literacy	109	26.7				
- Fair health literacy	168	41.1				
- Good health literacy	103	25.2				
- Excellent health literacy	29	7.1				
**Health behavior**			20.97	3.09	−0.241	<0.001 ***
- Poor health behavior	52	12.7				
- Fair health behavior	131	32				
- Good health behavior	127	31.1				
- Excellent health behavior	99	24.2				
**Total physical activity (min/ week)**			1306.00	1731.59	0.070	0.159
- Activity domain (min/week)			424.38	868.56	0.120	0.015 *
- Travel domain (min/week)			218.24	419.28	0.137	0.005 **
- Recreational domain (min/week)			663.37	1121.77	−0.037	0.460
PA ≥ 600 min/week	174	42.54				
PA < 600 min/week	235	57.46				
**Sedentary behavior**					−0.058	0.243
- Nonsedentary	60	14.7				
- Sometimes sedentary	203	49.6				
- Always sedentary	146	35.7				

* *p* < 0.05; ** *p* < 0.01; *** *p* < 0.001.

**Table 3 healthcare-10-00786-t003:** Stepwise multiple regression analysis model with stress level as the dependent variable.

Model	Independent Variables	R^2^	Adjusted R^2^	F	*p*-Value	Durbin–Watson
1	Self-Esteem	0.391	0.389	261.06	<0.001 ***	1.851
2	Self-Esteem, Study Period	0.417	0.414	145.02	<0.001 ***	
3	Self-Esteem, Study Period, Social Support	0.439	0.435	105.51	<0.001 ***	
4	Self-Esteem, Study Period, Social Support, Physical Activity (Travel Domain)	0.450	0.444	82.48	<0.001 ***	
5	Self-Esteem, Study Period, Social Support, Physical Activity (Travel Domain), Health Behavior	0.457	0.451	67.90	<0.001 ***	

*** *p* < 0.001.

**Table 4 healthcare-10-00786-t004:** Stepwise multiple regression of the relationship of self-esteem, study period, social support, physical activity in the travel domain, and health behavior with the stress level of Thailand university students during COVID-19.

Independent Variables	Constant	B	SE (B)	β	t	*p*-Value	Sr^2^ (Unique)	Tolerance	VIF
Self-Esteem	50.21	−0.680	0.058	−0.497	−11.664	<0.001 ***	−0.502	0.742	1.349
Study Period		−1.064	0.235	−0.168	−4.523	<0.001 ***	−0.220	0.976	1.024
Social Support		−0.084	0.023	−0.158	−3.621	<0.001 ***	−0.178	0.710	1.409
Physical Activity (Travel Domain)		0.002	0.001	0.108	2.900	0.004 **	0.143	0.975	1.026
Health Behavior		−0.199	0.083	−0.093	−2.390	0.017 *	−0.118	0.890	1.124

* *p* < 0.05; ** *p* < 0.01; *** *p* < 0.001.

## Data Availability

The datasets used and analyzed during this study are available from the corresponding author on reasonable request.

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
