# Peer review of "Psychosocial and Physical Predictors of Stress in University Students during the COVID-19 Pandemic: An Observational Study"

_healthcare, 2022, doi:10.3390/healthcare10050786_

Round 1

Reviewer 1 Report

General Comments

Thank you very much for giving me the opportunity to review the paper. The authors investigated the relationship between stress and psychosocial and physical factors among university students in Thailand. It is an important topic. However, there are several major issues.

Major Comments

This study is a cross-sectional study, also there is no control group. Therefore, the authours should not mention causality and the effect of COVID-19. In addition, it is necessary to describe in detail how to select research participants and what kind of selection bias exists. Also, the authors should discuss in detail the limitations of this study in the discussion section.

Author Response

Dear Editor and Reviewer 1,

Manuscript ID:         Healthcare-1642948

Manuscript Title:      Psychosocial and Physical Predictors of Stress in University Students during the COVID-19 Pandemic: An Observational Study

Comments from the Reviewer

Responses to the Reviewer

Thank you very much for giving me the opportunity to review the paper. The authors investigated the relationship between stress and psychosocial and physical factors among university students in Thailand. It is an important topic. However, there are several major issues.

We would like to thank the editor and reviewers for the thorough evaluation and review of our manuscript and for giving us the opportunity to make minor revisions to it. We truly value the comments and have made changes in the revised manuscript as recommended by the reviewers, on a point-by-point basis. Any revisions to the manuscript were marked up using the “Track Changes” function (as journal guide). The changes made are clearly shown in blue colored text below. We hope that our explanations and revisions would help clarify the issues raised by the reviewers. 

Point 1: This study is a cross-sectional study, also there is no control group. Therefore, the authors should not mention causality and the effect of COVID-19

We have made changes as suggested and included the following as a limitation of our study:

“Table 4. Stepwise multiple regression of the effects of relationship f self-esteem, year of study period, social support, physical activity in the travel domain, and health behavior, and gender with on stress level of Thailand university students during COVID-19.” (Page 8, lines 278-280).

“This study assessed the relationship of self-esteem, social support, health literacy and health behavior, and PA with stress in Thailand university students during the first wave of the COVID-19 pandemic.” (Page 8, lines 285–287).

“Our study has some limitations. First, it was a cross-sectional online survey that used a questionnaire to elicit information from participants. Therefore, there was no control group to determine a cause-and-effect relationship.” (Page 10, lines 378–385).

Point 2: In addition, it is necessary to describe in detail how to select research participants and what kind of selection bias exits.

We have added additional details on the selection of our participants as suggested. We have also stated clearly that  we used a probability sampling technique (proportional stratified sampling calculation) and the reasons for adopting this sampling technique.

“2.2. Participants

The sample size was calculated using the formula by Taro Yamane [17]. The population of students was 25,743. With a significance level of 0.05, the minimum valid sample size was calculated to be 395 students. Undertaking proportional stratified sampling of students from each faculty, 409 students at Burapha University were recruited for this study.

For confounders’ control, inclusion criteria were determined the student who are more than 18 years old, studying at the Bangsaen campus of Burapha University, and accessible in electronic devices for filling out the online questionnaires. Participants were excluded if they were unable to read or understand Thai language because the questionnaire was written in Thai. The confounders in the study were age, specific location, and nationality.

Participants were requested to participate in the study by responding to an online invitation posted on the official social media handles of each faculty’s student affairs (Facebook and LINE Application). The participants completed an anonymized online questionnaire that was created using Google Forms. Completed online questionnaire were collected and analyzed as participants of the study.” (Page 3, lines 97–117).

Point 3: Also, the authors should discuss in detail the limitations of this study in the discussion section

We have added more details on the limitations as suggested:

Our study has some limitations. First, it was a cross-sectional online survey that used a questionnaire to elicit information from participants. Therefore, there was no control group to determine a cause-and-effect relationship. In addition, the responses were mainly subjective, the questionnaire was subject to varied interpretation and understanding, and errors in responses and mental fatigue by participants. Second, depression, anxiety, and posttraumatic stress disorders, which have been linked to the COVID-19 pandemic, were not measured [36]. Future studies on these psychological disorders and other complications associated with the COVID-19 pandemic are needed.” (Page 10, lines 378–385).

Reviewer 2 Report

Dear authors, 

This study is an observational and analytical descriptive study. Please correct it in the 83th line. 

Your limitations is not few!! 

There are some the other limitations in your design such as accuracy of answers, Similar questions that students may not have understood the exact meaning of, mental fatigue, and ultimately errors in the answers provided can be other limitations of this study. You could active each link 10 minutes after filling the one questionnaire. 

"This study aimed to determine the effects of self-esteem, social support, health literacy and health behavior, and PA on the stress level of Thailand university students during the COVID-19 pandemic."

You can not claim that in this study you examined the causal relationship between the variables. Therefore, using the word "effect" in mentioning the research question is not correct.

Anxiety disorders such as phobias, panic and post-traumatic stress disorder are important limitations of your study. Because they can directly affect students' self-confidence. 

It is also important to control the other confounders in this study, which are not described. Using logistic regression can be helpful in this regard.

Please attach the questionnaires and mention the validity and reliability of them. 

Author Response

Dear Editor and Reviewer 2,  

Manuscript ID:         Healthcare-1642948

Manuscript Title:      Psychosocial and Physical Predictors of Stress in University Students

during the COVID-19 Pandemic: An Observational Study

We would like to thank the editor and reviewers for the thorough evaluation and review of our manuscript and for giving us the opportunity to make minor revisions to it. We truly value the comments and have made changes in the revised manuscript as recommended by the reviewers, on a point-by-point basis. Any revisions to the manuscript were marked up using the “Track Changes” function (as journal guide). The changes made are clearly shown in blue colored text below. We hope that our explanations and revisions would help clarify the issues raised by the reviewers. 

Comments from the Reviewer

Responses to the Reviewer

Point 1: This study is an observational and analytical descriptive study. Please correct it in the 83th line

We have made changes as suggested.

This was a cross-sectional, an observational and analytical descriptive study.” (Page 2, lines 92).

Point 2: There are some the other limitations in your design such as accuracy of answers, similar questions that students may not understood the exact meaning of, mental fatigue, and ultimate error in the answers provided can be other limitation of this study. You could active each link 10 minutes after filling the one questionnaire.

We have added more details about limitations as suggested. Additionally, we used only one link for the google form, which included a questionnaire composed of six sections which were clearly defined and segregated. We informed the participants to take a 5 mins rest  after completing each section of the questionnaire. We also revised participants section of manuscript.

“The participants were informed to rest for 5 mins after completing each section of the questionnaire.” (Page 3, lines 132-133).

“Our study has some limitations. First, it was a cross-sectional online survey that used a questionnaire to elicit information from participants. Therefore, there was no control group to determine a cause-and-effect relationship. In addition, the responses were mainly subjective, the questionnaire was subject to varied interpretation and understanding, and errors in responses and mental fatigue by participants. Second, depression, anxiety, and posttraumatic stress disorders, which have been linked to the COVID-19 pandemic, were not measured [36].” (Page 10, lines 378-384).

The online questionnaire link is as follows: https://docs.google.com/forms/d/e/1FAIpQLSd9T-D8jka9qUySsCSZoR5f1r_BJTvBIW2Oepio58tz_iE0Eg/viewform

Point 3: “This study aimed to determine the effects of self-esteem, social support, health literacy, and health behavior, and PA on the stress level of Thailand university students during the COVID-19 pandemic.” You can not claim that in this study you examined the causal relationship between the variables. Therefore, using the word “Effect” in mentioning the research question is not correct.

Thank you for your input, we have rephrased this accordingly to reflect that we only investigated the relationship between these factors and stress levels.

We revised discussion section of manuscript.

“This study assessed the relationship of self-esteem, social support, health literacy and health behavior, and PA with stress in Thailand university students during the first wave of the COVID-19 pandemic.” (Page 8, lines 285-287).

Point 4: Anxiety disorders such as phobias, panic, and post-traumatic stress disorder are important limitations of your study. Because they can directly affect student’s self confidence.

We already included this in the limitation:

“Second, depression, anxiety, and posttraumatic stress disorders, which have been linked to the COVID-19 pandemic, were not measured [36]. Future studies on these psychological disorders and other complications associated with the COVID-19 pandemic are needed.” (Page 10, lines 382-385).

Point 5: It is also important to control the other confounders in this study, which are not described. Using logistic regression can be helpful in this regard.

Thank you for your suggestion. This research has controlled the age of respondents to be more than 18 years old and studying in bachelor’s degree in Burapha University, specific location, nationality in order to control the confounders.

Additionally, we revised participant section of manuscript.

“For confounders’ control, inclusion criteria were determined the student who are more than 18 years old, studying at the Bangsaen campus of Burapha University, and accessible in electronic devices for filling out the online questionnaires. Participants were excluded if they were unable to read or understand Thai language because the questionnaire was written in Thai. The confounders in the study were age, specific location, nationality. (Page 3, lines 103-108).

We applied ordinary multiple regression technique because both independent and dependent variables are scale data. Then we applied ordinary multiple regression technique.
Additionally, we revised manuscript as presented in statistical analysis section of manuscript.

“Stress level (ratio level) was our dependent variable (DV) while our independent variables (IVs) were gender, age, study period, study program, self-esteem, social support, health literacy, health behavior, sedentary behavior, and PA (including the work, travel, and recreational domains). We performed ordinary multiple regression analysis because of its suitability to our data set. Our study’s nominal data (gender, study program [health sciences vs. nonhealth sciences], and sedentary behavior [nonsedentary vs. sedentary behavior]) were recoded and transformed into dummy variables.” (Page 4, lines 176-183).

Point 6: Please attach the questionnaires and mention the validity and reliability of them.

Please see the questionnaire through following links:

(1) Physical activity questionnaire: https://drive.google.com/file/d/1JdvXvAERc1yEc0q5Confwfm4kOmovxnC/view

Reference: World Health Organization. Global physical activity questionnaire (GPAQ) analysis guide. Available online: http://www.who.int/chp/steps/resources/GPAQ_Analysis_Guide.pdf.

(2) Social support questionnaire:

https://drive.google.com/file/d/1VfZTp7ZaO6ExywfMFGWXtv36W-MdIZqW/view

Reference: Wongpakaran, N.; Wongpakaran, T. A revised Thai multi-dimensional scale of perceived social support. Span. J. Psychol. 2012, 15, 1503-1509, doi:10.5209/rev_sjop.2012.v15.n3.39434.

(3) Self-esteem questionnaire: http://www.wongpakaran.com/images/column_1334240442/Rosenberg%20Self%20Esteem-revised_TNK.pdf

Reference: Wongpakaran, T.; Wongpakaran, N. A comparison of reliability and construct validity between the original and revised versions of the Rosenberg Self-Esteem Scale. Psychiatry Investig. 2012, 9, 54-58, doi:10.4306/pi.2012.9.1.54.

(4) Health Literacy and (5) health behavior questionnaire:

http://www.hed.go.th/linkhed/file/558

This questionnaire was the national survey questionnaire that was developed and approved by Ministry of Public Health, Thailand.

(6) Stress level questionnaire:

Reference: Wongpakaran, N.; Wongpakaran, T. The Thai version of the PSS-10: An Investigation of its psychometric properties. Biopsychosoc. Med. 2010, 4, 6, doi:10.1186/1751-0759-4-6.

In addition, the validity and reliability of the questionnaire has been included in the revised manuscript:

“PA was assessed using the Global Physical Activity Questionnaire (GPAQ) v.2 [18]. The Thai version of GPAQ was published by Thailand’s Ministry of Public Health. A previous study showed a reliability of 0.67 to 0.73 and a moderate (0.45) to strong (0.65) validity [19].” (Page 3, lines 134-137).

Levels of social support were measured using the Thai version of the revised Multidimensional Scale of Perceived Social Support.

“The reliability of this questionnaire was between 0.86 and 0.92 [21].” (Page 4, lines 149-150).

Self-esteem was measured using the Thai version of the Rosenberg Self-Esteem Scale.

“This questionnaire had a strong reliability of 0.86 [22].” (Page 4, lines 154-155).

Health literacy was assessed using the health literacy and health behaviors 3E 2S evaluation of the Health Education Division, of Thailand’s Ministry of Public Health.

“The Kuder–Richardson 20 score for this evaluation is 0.516 and it has Cronbach’s alpha coefficient of 0.602–0.788.” (Page 4, lines 161-162).

Stress levels were measured using the Thai version of the 10-item Perceived Stress Scale.

The reliability of this questionnaire was 0.85 [23]. (Page 4, lines 166).

Reviewer 3 Report

My review report for healthcare-1642948

I appreciate the hard work of the authors for carrying out a detailed assessment regarding the effect of recent pandemic on the mental and physical aspect of University students. Though the study is very interesting, it needs to clarify some important issues. I recommend a major revision before publication.

Specific comments to the authors:

  1. In the introduction section, the authors presented several arguments regarding different psychological factors. They showed that absence of these brought negative consequences. Therefore, it is already established. What is the new fact the authors have introduced through this study?
  2. It is not clear why the authors used stratified sampling. How they determined the minimum sample size? Why sample size 409? Why the other participants not familiar with Thai language were excluded? It would be more interesting to see and make a comparison between the level of stress among the local students and the outsiders.
  3. “The contents of each section were from standardized questionnaires”- needs explanation. Please provide the links of the questionnaire and your final questionnaire.
  4. The data is very old i.e. from October 2020. However, it was noticed that the pandemic continued intensely during most of the 2021 as well. In South-East Asia, the human beings were affected most during the second wave in the year 2021. Following this, why did the authors not update their data?      
  5. Why different sets of likert scales were used? There is a possibility that higher likert scale values will obtain more weightage.
  6. Further, this study did not justify the use of ordinary regression analysis and correlation. Why not spatial lag and error model or geographically weighted regression or Moran’s I?
  7. The result section is written very poor. No proper analysis and explanation is provided regarding the outputs of the tables.
  8. In the discussion section, the recommendations made by the authors were very general. This needs to be more specific and objective oriented.

      9. The conclusion section needs to be extended further by including the overall summary of the research, important contribution, and future research direction.

Author Response

Dear Editor and Reviewer 3,

Manuscript ID:         Healthcare-1642948

Manuscript Title:      Psychosocial and Physical Predictors of Stress in University Students

during the COVID-19 Pandemic: An Observational Study

Comments from the Reviewer

Responses to the Reviewer

I appreciate the hard work of the authors for carrying out a detailed assessment regarding the effect of recent pandemic on the mental and physical aspect of University students. Though the study is very interesting, it needs to clarify some important issues. I recommend a major revision before publication.

We would like to thank the editor and reviewers for the thorough evaluation and review of our manuscript and for giving us the opportunity to make minor revisions to it. We truly value the comments and have made changes in the revised manuscript as recommended by the reviewers, on a point-by-point basis. Any revisions to the manuscript were marked up using the “Track Changes” function (as journal guide). The changes made are clearly shown in blue colored text below. We hope that our explanations and revisions would help clarify the issues raised by the reviewers. 

Point 1: In the introduction section, the authors presented several arguments regarding different psychological factors. They showed that absence of these brought negative consequences. Therefore, it is already established. What is the new fact the authors have introduced through this study?

Thank you very much for this valuable contribution. We have clarified this to show the importance of our study.

“COVID-19 increases stress in students, activating physiological, emotional, and behavioral responses [14]. Moreover, different situations, areas and occupations could influence the impact of the pandemic on individuals and population categories. There is no study on the impact of the pandemic, especially the first wave, on Thailand university students. Due to the novelty of the pandemic and accompanying restrictions, Thai university students are at high risk of stress because of learning adjustment, educational interruption, financial concerns, family problem, alterations in daily life, social isolation, etc This study investigated the psychosocial and physical factors influencing the stress level in Thailand university students during the COVID-19 pandemic. Understanding these factors can be beneficial in the development of strategies aimed at stress reduction and prevention of psychological disorders.” (Page 2, lines 79-89).

Point 2: It is not clear why the authors used stratified sampling. How they determined the minimum sample size? Why sample size 409? Why the other participants not familiar with Thai language were excluded? It would be more interesting to see and make a comparison between the level of stress among the local students and the outsiders.

We used stratified random sampling due to the differences in the characteristics (heterogeneity) of each faculty. The questionnaire was in Thai language, hence we excluded those who were not familiar with the language. We have also explained how we arrived at our sample size.

“The sample size was calculated using the formula by Taro Yamane [17]. The population of students was 25,743. With a significance level of 0.05, the minimum valid sample size was calculated to be 395 students. Undertaking proportional stratified sampling of students from each faculty, 409 students at Burapha University were recruited for this study. (Page 3, lines 98-102).

“Participants were excluded if they were unable to read or understand Thai language because the questionnaire was written in Thai.” (Page 3, lines 105-107).

We agree with your valuable suggestion that it would have been more interesting to see and make a comparison between the level of stress among local students and the outsiders. This was however not assessed in this study. However, we have included in our conclusion that it is worth exploring in future studies:

“Further studies should investigate the long-term effect of COVID-19 pandemic on stress level and the stress management strategies among specific categories of university students. Moreover, experimental studies are needed to confirm the positive effect of self-esteem and social support on stress levels during pandemics or life-threatening situations.” (Page 10, lines 399-402).

Point 3: “The contents of each section were from standardized questionnaires”- needs explanation. Please provide the links of the questionnaire and your final questionnaire.

We have made changes to this section as suggested and also provided the links to the questionnaires below:

“The validity and reliability of the instrument was assured through wide consultations with relevant academicians, review of literature and adaptation of validated instruments.” (Page 3, lines 128-130).

Links to the original questionnaire are as follows:

(1) Physical activity questionnaire: https://drive.google.com/file/d/1JdvXvAERc1yEc0q5Confwfm4kOmovxnC/view

Reference: World Health Organization. Global physical activity questionnaire (GPAQ) analysis guide. Available online: http://www.who.int/chp/steps/resources/GPAQ_Analysis_Guide.pdf.

(2) Social support questionnaire:

https://drive.google.com/file/d/1VfZTp7ZaO6ExywfMFGWXtv36W-MdIZqW/view

Reference: Wongpakaran, N.; Wongpakaran, T. A revised Thai multi-dimensional scale of perceived social support. Span. J. Psychol. 2012, 15, 1503-1509, doi:10.5209/rev_sjop.2012.v15.n3.39434.

 (3) Self-esteem questionnaire: http://www.wongpakaran.com/images/column_1334240442/Rosenberg%20Self%20Esteem-revised_TNK.pdf

Reference: Wongpakaran, T.; Wongpakaran, N. A comparison of reliability and construct validity between the original and revised versions of the Rosenberg Self-Esteem Scale. Psychiatry Investig. 2012, 9, 54-58, doi:10.4306/pi.2012.9.1.54.

(4) Health Literacy and (5) health behavior questionnaire:

http://www.hed.go.th/linkhed/file/558

This questionnaire was the national survey questionnaire that was developed and approved by Ministry of Public Health, Thailand.

(6) Stress level questionnaire:

Reference: Wongpakaran, N.; Wongpakaran, T. The Thai version of the PSS-10: An Investigation of its psychometric properties. Biopsychosoc. Med. 2010, 4, 6, doi:10.1186/1751-0759-4-6.

In addition, the validity and reliability of the questionnaire has been included in the revised manuscript:

“PA was assessed using the Global Physical Activity Questionnaire (GPAQ) v.2 [18]. The Thai version of GPAQ was published by Thailand’s Ministry of Public Health. A previous study showed a reliability of 0.67 to 0.73 and a moderate (0.45) to strong (0.65) validity [19].” (Page 3, lines 134-137).

Levels of social support were measured using the Thai version of the revised Multidimensional Scale of Perceived Social Support.

“The reliability of this questionnaire was between 0.86 and 0.92 [21].” (Page 4, lines 149-150).

Self-esteem was measured using the Thai version of the Rosenberg Self-Esteem Scale.

“This questionnaire had a strong reliability of 0.86 [22].” (Page 4, lines 154-155).

Health literacy was assessed using the health literacy and health behaviors 3E 2S evaluation of the Health Education Division, of Thailand’s Ministry of Public Health.

“The Kuder–Richardson 20 score for this evaluation is 0.516 and it has Cronbach’s alpha coefficient of 0.602–0.788.” (Page 4, lines 161-162).

Stress levels were measured using the Thai version of the 10-item Perceived Stress Scale.

The reliability of this questionnaire was 0.85 [23]. (Page 4, lines 166).

The link to our final questionnaire: https://docs.google.com/forms/d/e/1FAIpQLSd9T-D8jka9qUySsCSZoR5f1r_BJTvBIW2Oepio58tz_iE0Eg/viewform

Point 4: The data is very old i.e. from October 2020. However, it was noticed that the pandemic continued intensely during most of the 2021 as well. In South-East Asia, the human beings were affected most during the second wave in the year 2021. Following this, why did the authors not update their data? 

Actually, this paper was submitted to another journal in last year. It was suspended for almost a year. Finally, we were informed that our paper is not in line with the journal’s scope even though we had investigated the scope of the journal carefully.

You indeed made a valid point. However, we would like to advance another view point, which is that every wave of COVID-19 can have different impacts on human beings, which makes each wave important and worth exploring. Thus, our study focused on the first wave of COVID-19 in Thailand, a situation that has not been studied and published, even as at this current year, 2022. Thus, despite focusing on the first wave, we believe that our study is relevant and the findings useful for future pandemic.

Additionally, we have mentioned this view points in introduction and discussion sections:

“There is no study on the impact of the pandemic, especially the first wave, on Thailand university students. Due to the novelty of the pandemic and accompanying restrictions, Thai university students are at high risk of stress because of learning adjustment, educational interruption, financial concerns, family problem, alterations in daily life, social isolation, etc This study investigated the psychosocial and physical factors influencing the stress level in Thailand university students during the COVID-19 pandemic. Understanding these factors can be beneficial in the development of strategies aimed at stress reduction and prevention of psychological disorders” (Page 3, lines 81-89).

 “This study provided knowledge on stress level related to psychosocial and physical factors during the first wave of the COVID-19 pandemic. Our result can be applied for the proper management of new and life-threatening pandemic, which impact human well-being.” (Page 10, lines 370-373).

Point 5: Why different sets of likert scales were used? There is a possibility that higher likert scale  values will obtain more weightage.

Based on the recommendation of our statistician consultants, we would like to explain as follows:

The instrument in the research is the standard instrument applied in Thai academician. The scale is not adjusted in order to maintain the reliability of the instrument.

Your concern is valid. There is higher possibility to have higher weightage on higher Likert scale. Its effect was performed in coefficient level (B) on table 4. We can use the equation to justify the stress level. If you would like to compare the sequence of factors, standardized coefficient is recommended (table 4).   

Additionally, we have added more details in the Statistical analysis section

“To maintain the reliability of the instrument and reduce complexity in applying the model, the scale was not adjusted in this study. Due to concerns about the variability of different weightage Likert scale, the unstandardized coefficient was reported to predict the stress level. The model can be implemented into unobserved observation without transforming the data. Additionally, the standardized coefficient was reported to account for the comparison on the strength of factors affecting stress level.” (Page 4, lines 183-188).

Point 6: Further, this study did not justify the use of ordinary regression analysis and correlation. Why not spatial lag and error model or geographically weighted regression or Moran’s I?

Due to the limitation of statistical software, the geographically weighted regression or Moran’s I could not be applied. We conducted statistical tests with standardized data. However, we used Pearson’s correlation to determine correlation instead. The reason we used ordinary regression analysis is to justify the effect between gender, age, study period, study program, self-esteem, social support, health literacy, health behavior, sedentary behavior, PA, and stress level because the data are all scale data which is appropriate and sufficient to apply instead of using geographically weighted regression.

Additionally, we have added more details in the Statistical analysis section:

“Data was analyzed using SPSS software. Stress level (ratio level) was our dependent variable (DV) while our independent variables (IVs) were gender, age, study period, study program, self-esteem, social support, health literacy, health behavior, sedentary behavior, and PA (including the work, travel, and recreational domains). We performed ordinary multiple regression analysis because of its suitability to our data set. Our study’s nominal data (gender, study program [health sciences vs. nonhealth sciences], and sedentary behavior [nonsedentary vs. sedentary behavior]) were recoded and transformed into dummy variables. To maintain the reliability of the instrument and reduce complexity in applying the model, the scale was not adjusted in this study. Due to concerns about the variability of different weightage Likert scale, the unstandardized coefficient was reported to predict the stress level. The model can be implemented into unobserved observation without transforming the data. Additionally, the standardized coefficient was reported to account for the comparison on the strength of factors affecting stress level. Furthermore, assumptions of ordinary regression analysis were evaluated. The Kolmogorov–Smirnov test and scatter plots were used to test the normality and homoscedasticity of residuals. Autocorrelation and the multicollinearity of IVs were tested using Durbin–Watson statistic, tolerance test, and variance inflation factor. Pearson’s product–moment correlation coefficient was used to examine the relationship between the stress level and demographic variables, as well as between the stress level and DVs. The significance level in the study was 0.05.” (Page 4, lines 175-195).

Point 7: The result section is written very poor. No proper analysis and explanation is provided regarding the outputs of the tables.

Thank you for your feedback. We have reworked the result section for clarity and detail.

 “A total of 523 questionnaires were retrieved form the students, however, 409 completed questionnaires were analyzed and included in this result. Participants’ characteristics are presented in Table 1. Majority of the participants were females ( ) and in nonhealth sciences programs ( ) while about 40% ( ) were in the fourth academic year.

The correlations between the participants’ characteristics and stress level are presented on Table 1. Age ( ) and study period ( ) had a weak, significant negative correlation with stress level.

The correlations between independent variables and stress level are presented on Table 2. Self-esteem ( ) and social support ( ) had a moderate, significant negative correlation with stress level. On the other hand, health literacy ( ) and health behavior ( ) had a weak, significant negative correlation with stress level. Activity domain ( ) and travel domain ( ) of PA had a weak, significant positive correlation with stress level. [24]. The remaining variables were not significant.

The results of ordinary multiple regression analysis are presented in Table 3. The variables that were not significant at correlation were excluded from this analysis. Stepwise method iteratively examines each significant independent variable in the model. The regression model comprised of five significant independent variables (F = 67.90, p < 0.001): self-esteem, study duration, social support, the travel domain of PA, and health behavior. Overall, 45.7% of the variability in stress level was predicted by these five independent variables. In addition, the Durbin Watson test of autocorrelation ( ) result was 1.851. Based on the acceptable range of no evidence of autocorrelation (1.794 and 2.206), our result showed that there is no evidence of autocorrelation. The residuals presented normality and homoscedasticity.

In this study, multicollinearity was also investigated by Variable Inflation Factors (VIF), which should be close to 1 and lower than 5. As shown on Table 4, the VIF value for each variable is between 1.024 and 1.409. This finding shows that our study had no multicollinearity issue.

Table 4 shows the unstandardized regression coefficients ( ), the standardized regression coefficient (β) and the semi-partial correlations (Sr2). For the final model, five IVs contributed significantly to stress level prediction: self-esteem ( ), study period ( ), social support ( ), the travel domain of PA ( ), and health behavior ( ). Hence, equation model for predicting stress level was derived for these variables:

 Predicted stress level = 50.21 − 0.68 (self-esteem) − 1.064 (study period) − 0.084 (social support) + 0.002 (travel domain of PA) − 0.199 (health behavior)”

Page 4-5, lines 196-235).

Point 8: In the discussion section, the recommendations made by the authors were very general. This needs to be more specific and objective oriented.

Thank you for your feedback, we have reviewed the discussion and made study specific recommendations. We have more highlighted our objective oriented in introduction section and recommendation in discussion section as suggested.

“There is no study on the impact of the pandemic, especially the first wave, on Thailand university students. Due to the novelty of the pandemic and accompanying restrictions, Thai university students are at high risk of stress because of learning adjustment, educational interruption, financial concerns, family problem, alterations in daily life, social isolation, etc This study investigated the psychosocial and physical factors influencing the stress level in Thailand university students during the COVID-19 pandemic. Understanding these factors can be beneficial in the development of strategies aimed at stress reduction and prevention of psychological disorders” (Page 3, lines 81-89).

“An interesting finding of this study was that, during the COVID-19 pandemic, there was an increase in travel activity among students and this was positively associated with high levels of stress. During first wave of the COVID-19 pandemic in Thailand, most of the universities adopted online learning from home for their students. They have the least medical risks that allows them to journey outside for getting necessities for a living to support their family. In normal situations, individuals who are regularly physically active have lower levels of stress than those who are less active [1]. The higher stress level during travel is likely to be due to feared exposure to COVID-19. Although the students will have been wearing masks and social distancing, there is still a risk of COVID-19 infection during travel to public areas. Moreover, the low confidence in pandemic control and the lack of resources for fighting COVID-19 might further affect stress level in these circumstances, especially the first wave of pandemic [25,34].” (Page 9, lines 337-348).

“This study provided knowledge on stress level related to psychosocial and physical factors during the first wave of the COVID-19 pandemic. Our result can be applied for the proper management of new and life-threatening pandemic, which impact human well-being.” (Page 10, lines 370-373).

Point 9: The conclusion section needs to be extended further by including the overall summary of the research, important contribution, and future research direction.

We have made changes to the conclusion as suggested.

“The first wave of COVID-19 pandemic negatively impacted on the stress levels of university students in Thailand. Majority of them presented with moderate stress level. The combination of psychosocial and physical factors including self-esteem, study duration, social support, health behavior, and the travel domain of PA predicted their level of stress during the pandemic. Having a greater level of self-esteem and social support were significant stress coping strategies for the students. On the other hand, students experienced higher stress with travels within the public space. Further studies should investigate the long-term effect of COVID-19 pandemic on stress level and the stress management strategies among specific categories of university students. Moreover, experimental studies are needed to confirm the positive effect of self-esteem and social support on stress levels during pandemics or life-threatening situations.” (Page 10, lines 392-402).

Please use the link in attachment 
